# Research Progress on the Predicting Factors and Coping Strategies for Postoperative Recurrence of Esophageal Cancer

**DOI:** 10.3390/cells12010114

**Published:** 2022-12-28

**Authors:** Yujie Zhang, Yuxin Zhang, Lin Peng, Li Zhang

**Affiliations:** 1Department of Oncology, Tongji Medical College, Tongji Hospital, Huazhong University of Science and Technology, No. 1095 Jiefang Avenue, Wuhan 430030, China; 2Department of Pediatric Surgery, Tongji Medical College, Tongji Hospital, Huazhong University of Science and Technology, No. 1095 Jiefang Avenue, Wuhan 430030, China

**Keywords:** esophageal cancer, postoperative recurrence, metastasis, clinical biomarkers, molecular markers

## Abstract

Esophageal cancer is one of the malignant tumors with poor prognosis in China. Currently, the treatment of esophageal cancer is still based on surgery, especially in early and mid-stage patients, to achieve the goal of radical cure. However, esophageal cancer is a kind of tumor with a high risk of recurrence and metastasis, and locoregional recurrence and distant metastasis are the leading causes of death after surgery. Although multimodal comprehensive treatment has advanced in recent years, the prediction, prevention and treatment of postoperative recurrence and metastasis of esophageal cancer are still unsatisfactory. How to reduce recurrence and metastasis in patients after surgery remains an urgent problem to be solved. Given the clinical demand for early detection of postoperative recurrence of esophageal cancer, clinical and basic research aiming to meet this demand has been a hot topic, and progress has been observed in recent years. Therefore, this article reviews the research progress on the factors that influence and predict postoperative recurrence of esophageal cancer, hoping to provide new research directions and treatment strategies for clinical practice.

## 1. Introduction 

Esophageal cancer (EC) is a malignant tumor with high morbidity and mortality, and about 90% of the pathological subtypes of esophageal cancer in China are squamous cell carcinoma [1,2]. At present, the primary treatment modality of esophageal cancer is still comprehensive surgery-based treatment [3]. In recent years, the continuous progress of minimally invasive surgical techniques, the constant development of radiotherapy techniques and equipment, as well as the emergence of new chemotherapy and immunotherapy drugs significantly improved the survival and prognosis of esophageal cancer patients. However, the prospect for overall survival of esophageal cancer patients is still not optimistic [4]. Esophageal cancer is a malignant tumor with high recurrence and metastasis. After radical resection, the total recurrence rate is as high as 27~52.4%, and the local recurrence rate is as high as 32.6~49%. After radical resection, the distant metastasis rate of patients with positive lymph nodes is as high as 19.8~61.3%, which is one of the main reasons for treatment failure and poor prognosis of esophageal cancer patients [5,6,7].

Currently, the clinical prediction, prevention and treatment of postoperative recurrence of esophageal cancer are not satisfactory. Regular postoperative monitoring and follow-up are essential for the early detection of recurrence and metastasis in esophageal cancer patients. Identification of risk factors for early recurrence after surgery is therefore crucial, as patients can be found to have recurrent lesions and be treated early, which improves the prognosis of the patients. The main purpose of this paper is to review the recurrence model and risk factors of esophageal cancer from three aspects—relevant clinical factors influencing recurrence, characteristic parameters predicted by imageomics, and molecular markers—in order to provide references for postoperative surveillance and follow-up strategies of esophageal cancer.

## 2. Relevant Clinical Factors Influencing Recurrence

### 2.1. The Degree of Tumor Differentiation

The degree of tumor differentiation, which is histological grade, has a close relationship with the biological behavior of tumors. It is an indicator that reflects the degree of malignancy, and is closely related to the prognosis of esophageal cancer [8]. In a clinical study that enrolled 61 patients with postoperative recurrence of esophageal cancer, researchers found that poor tumor differentiation was an independent risk factor for postoperative recurrence in patients [9]. Zhang et al. [10] reported a study of 408 patients with recurrence after esophagectomy and local lymph node dissection. Univariate and multivariate analysis showed that the degree of tumor differentiation was significantly associated with distant metastasis, which was an independent risk factor for postoperative recurrence. 

Thomson et al. [11] reported similar results in their retrospective study that enrolled 221 esophageal cancer patients who had undergone esophagectomy without neoadjuvant or adjuvant therapy, and found that poor tumor differentiation was associated with postoperative distant metastasis. In a study of 149 patients with recurrence after esophagectomy by Hulscher et al. [12], the results of multivariate analysis showed that the low degree of tumor differentiation was independently associated with the risk of recurrence and metastasis after esophagectomy. Gertler et al. [13] found that after the initial surgery in esophageal cancer patients, the degree of tumor differentiation was an independent predictive parameter for the risk of lymph node metastasis, as shown by the pathological analysis. Stiles et al. suggested that in patients with ESCC treated with neoadjuvant therapy combined with surgery, the low degree of tumor differentiation is closely related to the poor prognosis [14]. The above research results on the risk factors of postoperative recurrence of esophageal cancer are consistent. Poorly differentiated and undifferentiated tumors are highly capable of progression. Even when no evident cancer is found during the operation, there may have been local or distant subclinical metastasis, leading to postoperative recurrence or metastasis and affecting the prognosis of the patient. 

To summarize, a large number of retrospective studies have shown that the degree of tumor differentiation may be an independent risk factor for postoperative recurrence in esophageal cancer patients, which can predict the risk probability of postoperative recurrence of esophageal cancer patients. 

### 2.2. TNM Stage and Other Tumor Characteristics

The TNM stage and other tumor characteristics, including tumor length and vascular invasion, are inextricably associated with tumor recurrence.

The T stage refers to the condition of the primary tumor and is an important basis for the pathological staging of esophageal cancer [15]. According to a study, tumor histological differentiation grade, tumor length and advanced clinical T stage were related to occult lymph node metastasis in clinical T1 to T2 N0 cancer [16]. A retrospective study of 582 patients with ESCC who had undergone surgical resection showed that the length of the tumor reflected the longitudinal spread of cancer cells, which may affect the survival prognosis of patients. The 5-year survival rates of patients with tumor lengths of 1 cm, 2 cm, 3 cm and more than 3 cm were 77.3%, 48.1%, 38.5% and 23.3%, respectively [17]. Similarly, a study showed that the tumor length was related to the survival of patients with esophageal cancer—that is, when the tumor length was less than 3 cm, the median overall survival was 47.1 months; when the length was between 3 cm and 4.4 cm, the median overall survival was 19.6 months; and when the length was greater than 4.5 cm, the median overall survival was 18 months [18]. Yoshida et al. [19] retrospectively analyzed the clinical data of 128 patients with esophageal cancer who had undergone radical surgery after neoadjuvant radiotherapy and chemotherapy, and found that pathological vascular invasion was one of the high-risk factors for early postoperative recurrence. Zhu et al. [20] identified the depth of tumor invasion and tumor margin as valuable predictors of early recurrence and poor prognosis in patients with ESCC. Many studies have suggested that the tumor margin status can be used as a predictor of postoperative survival of esophageal cancer patients [21,22]. In a retrospective analysis of 329 patients who had undergone esophagectomy, the presence of small tumors at or within 1 mm of the resection edge was not an important prognostic factor [23]. However, the study of Dexter et al. [24] proposed that the presence of tumor cells within 1 mm of the resection edge of esophageal cancer was an independent predictor of poor prognosis in patients with esophageal cancer. Therefore, it is important to evaluate the size of the primary esophageal tumor, depth of invasion and surgical margin. These measures are key predictors of recurrence, metastasis and prognosis of esophageal cancer.

The N stage describes the regional lymph node involvement. Lymph node metastasis is one of the main ways for tumors to metastasize and a key predictor of recurrence, metastasis and poor prognosis of esophageal cancer. Some studies demonstrated that with the increase in the number of positive lymph nodes came the decrease in the survival rate of esophageal cancer patients [25], while others showed that when the number of involved lymph nodes exceeded a critical point, the prognosis of patients was poor. In a study that included 536 patients who had undergone neoadjuvant chemoradiotherapy combined with surgery for esophageal cancer [26], the number of metastatic lymph nodes (≤4 or >4) and the proportion of metastatic lymph nodes of the dissected ones (≤0.2 or >0.2) were shown to be important predictors of patient outcomes. When the number of metastatic lymph nodes was more than 4, the 5-year survival rate of patients was only about 8%. When the proportion of metastatic lymph nodes among the lymph nodes submitted for examination was more than 0.2, the 5-year survival rate was about 22%. This may be because, in the early stage of cancer, tumor cells metastasize through lymphatic reflux. In a study that enrolled 61 patients with recurrence after neoadjuvant chemoradiotherapy combined with surgery for esophageal cancer (43 adenocarcinomas and 18 squamous carcinomas), patients with esophageal adenocarcinoma were found to have a higher incidence of distant recurrence, and patients with ESCC had a higher incidence of local recurrence. Given that the most common site of local recurrence in patients with ESCC is the mediastinal lymph node, more thorough or extensive mediastinal lymphadenectomy during surgery may contribute to a lower recurrence rate postoperatively. Regional lymph node involvement is an independent risk factor for survival and recurrence after surgery for esophageal cancer [27], and the number of involved lymph nodes is implicated in predicting early postoperative recurrence and death in patients with esophageal cancer. However, the impact of different numbers of positive regional lymph nodes on patient survival is variable. In clinical practice, further stratification may be performed based on the number of involved lymph nodes to assess the recurrence and prognosis of patients.

Although the tumor and positive lymph nodes can be resected to the greatest extent, it is not curative for micrometastases [28,29]. With the proliferation and lymph node metastasis of residual tumors, the TNM stage of tumors gradually changes, and the change in TNM stage is one of the main risk factors for postoperative recurrence of esophageal cancer patients. Hamai et al. [30], in their retrospective analysis of clinical data from 141 patients with ESCC who had undergone curative surgery after neoadjuvant chemoradiotherapy, revealed that the T stage (2/3/4) and N stage (2/3) after neoadjuvant therapy were significantly correlated with early postoperative recurrence, while the T stage (2/3/4) and N stage (2/3) after surgery were significantly correlated with poor postoperative prognosis. 

The death of esophageal cancer patients undergoing surgical treatment is mainly caused by local recurrence or metastasis of the tumor. Therefore, patients undergoing radical surgery should be followed up with regularly, including clinical reexamination and monitoring of the residual concealed micrometastasis and recurrence with sensitive detection methods, and timely treatment should be given to the patients, which is expected to improve the prognosis of patients. The primary tumor and lymph node metastasis of esophageal cancer reflect the malignant degree of the tumor. At present, the TNM stage of esophageal cancer mainly predicts the recurrence and prognosis of patients from the malignant biological behavior of the tumor. 

## 3. Imaging Omics for the Prediction of Stage, Response to Treatment and Prognosis in ESCC

Although imaging diagnostic techniques for esophageal cancer have been developing rapidly over the years, many limitations still remain for their applications, such as the characterization of target lesions, tumor staging, monitoring of treatment effects and prediction of survival prognosis. Imaging omics, as a novel individualized precision medicine technology, improves the clinical application of medical imaging limitations by transforming the regions of interest in medical images into image feature data through algorithms and performing quantitative analysis to obtain comprehensive feature information of tumors [31]. Radiomics extracts a lot of tumor information through the post-processing of medical images, which incorporates assessment features of differences in tumor biological behavior or intratumor heterogeneity, that is, alterations in the intratumoral microenvironment that can be visualized on images [32]. The spatial heterogeneity of tumors is caused by multifaceted factors, such as metabolism, vascularity, hypoxia, gene regulation and expression differences, and is usually closely associated with poor patient prognosis [33,34,35]. Therefore, analysis of the tumor-related information extracted through medical imaging can fill the gaps left by conventional clinical evaluation indicators to help clarify the tumor stage of patients, monitor the response and assess the prognosis [31,36,37,38,39]. 

Traditional imaging examination has some limitations in tumor staging. For the staging of esophageal cancer, Yang et al. retrospectively analyzed the preoperative contrast-enhanced CT images of 116 patients with ESCC who had received esophagectomy. The study suggested that CT contrast radiomics features were significantly related to T stage and tumor length, and showed favorable predictive performance for predicting preoperative pathological T stage and tumor length in ESCC patients [40]. Liu et al. [41] extracted the texture parameters of preoperative CT images of 73 patients with ESCC to study the correlation between the parameters and tumor stage. It was found that the imaging parameters extracted from the contrast-enhanced CT images showed good performance in the T, N and total staging. Accurate staging and prediction of lymph node metastasis based on imaging omics analysis can reduce the false negative rate of lymph node dissection, which is essential for treatment decisions and prognostic evaluation of patients. Qu et al. [42] analyzed 181 patients undergoing radical resection of esophageal cancer, including 90 in the training group and 91 in the test group. A prediction model was established by combining the pre-operative MRI image texture with the lymph node metastasis from the post-operative pathological results. The study found that the model had significant predictive performance, with potential to predict whether lymph nodes metastasized. Shen et al. [43] studied the preoperative CT imaging parameters of 197 patients with esophageal cancer, and constructed and tested the prediction model of lymph node metastasis. The results suggested that the model has good predictive power in the training group and the test group, and can be used to predict the preoperative lymph node metastasis of esophageal cancer patients. In a retrospective study conducted by Tan et al. [44], 154 patients with ESCC who had undergone radical resection were divided into a training group and a test group of 76 patients. The characteristic parameters of imageomics extracted from the arterial phase CT images of preoperative tumors were combined with the lymph node status detected by preoperative CT imaging to establish an individualized prediction model. It was found that the individualized prediction model had a good prediction effect in the training group and the test group. The researchers noted that the individualized prediction model provided accurate prediction of lymph node metastasis for patients with ESCC before treatment, and its prediction effect was better than clinical judgment. Dong et al. [45] retrospectively reviewed the preoperative PET/CT images of 40 patients with ESCC who had received surgical treatment. By analyzing the relationship between the texture characteristics of the images and the maximum standard uptake value (SUVmax), tumor pathological grade, tumor location and TNM stage, it was found that the imaging parameters were closely related to the tumor T and N stages, and could accurately identify tumors above stage IIB. In terms of therapeutic efficacy and prognosis prediction, Connie Yip and colleagues [46] retrospectively analyzed the CT images of 36 patients with esophageal cancer undergoing radiotherapy and chemotherapy, and evaluated the prognosis using image texture features. The results demonstrated that the CT texture parameters after treatment were correlated with the survival time, and the combination of the CT texture parameters before treatment with the thickness of the tumor predicted prognosis better than CT images after treatment alone. In another study [47], the enhanced CT images of 31 patients with esophageal cancer undergoing neoadjuvant chemotherapy before and after chemotherapy were compared and analyzed. It was found that the tumor texture became uniform after treatment, and multivariate analysis showed that the characteristic parameters could predict the treatment effect of chemotherapy, and the changes in the characteristic parameters could evaluate the prognosis difference of patients. Ganeshan et al. [32] compared the SUV value and texture feature extracted from the PET/CT images of 21 patients with esophageal cancer in the evaluation of survival and prognosis. The analysis found that the texture feature of CT images was superior to the SUV value in predicting the prognosis of esophageal cancer. Junfeng Xiong and colleagues [48] included 30 patients with esophageal cancer in their study. They extracted the characteristic imaging omics parameters from the PET images of the patients before and during the treatment of concurrent radiotherapy and chemotherapy, and built a prediction model in combination with the clinical parameters. The results demonstrated that the accuracy and specificity of the model in predicting the local control rate of tumors were above 90%, and the sensitivity was about 85%. This model can differentiate between patients with different risks of local cancer control failure after concurrent radiotherapy and chemotherapy for esophageal cancer, and may help to provide personalized treatment to patients. In the retrospective study of Larue et al. [49], 239 patients with esophageal cancer who had been treated with concurrent radiotherapy and chemotherapy after operation were included, and the imaging characteristics of CT images before treatment were obtained. The model for predicting survival was established based on the three-year overall survival rate of the patients. The results suggested that the predictive ability of the model for prognosis was superior to that of using standard clinical indicators alone. Qiu et al. evaluated 206 ESCC patients who achieved pathological complete response after neoadjuvant chemoradiotherapy followed by surgery, including 146 in the training cohort and 60 in the validation cohort. The study developed a radiomics nomogram model incorporating eight radiomics features and clinical factors. This model has the improved ability to predict the postoperative recurrence risk in the studied patients [50]. 

Since the first application of imaging omics in esophageal cancer research, the progress of imaging omics in esophageal cancer has been rapidly advancing, showing excellent performance in various aspects, such as qualitative diagnosis, clinical staging, efficacy assessment, and prognosis prediction, and accelerating the process of individualization of esophageal cancer treatment. Its advantage lies in the noninvasive approach, as well as the direct utilization of information extracted from existing images for analysis. However, there are also many shortcomings of imaging omics in esophageal cancer. For example, most of the current studies are exploratory, so the repeatability of the results is poor and the level of evidence is low. The rapid development of medical imaging technology and unified standardized research parameters of imaging omics will bring a promising prospect to the application of imaging omics in predicting recurrence after treatment of esophageal cancer.

## 4. Predictive Molecular Markers of Postoperative Recurrence and Metastasis

Biomarkers are characteristic indicators in the peripheral blood, tissues or cells of patients that can reflect the pathological process of tumors, and are used for screening, early diagnosis and response evaluation of tumors [51,52]. Changes in the expression of biomarkers can help detect malignant tumors, monitor recurrence and metastasis or evaluate the efficacy and prognosis before lesions can be captured by imaging. They can also be used to develop new targeted therapeutic drugs. The identification and validation of biomarkers are crucial to the formulation of effective screening and examination methods, which helps to detect the recurrence of ESCC after treatment as early as possible and improve the survival of patients [53]. An ideal biomarker should be highly sensitive and specific, and can be used for qualitative, localization and differential diagnosis of cancer. However, no specific biomarker for ESCC has been found at present [54], and the need for early detection of recurrence and metastasis cannot be realized. It is urgent to explore new biomarkers.

### 4.1. Blood Molecular Biomarkers

Blood biomarkers have attracted wide attention due to their advantages such as simplicity, short-term repeatability and economy. They are valuable in assisting the diagnosis of tumors and improving the prognosis of patients. Measurement of tumor markers in peripheral blood can determine the expression levels and the threshold values for tumors. It has certain clinical value in tumor screening, diagnosis, pathological analysis, response evaluation, monitoring of recurrence or metastasis and prediction of prognosis.

#### 4.1.1. Clinical Tumor Markers

The commonly used tumor markers in clinical diagnosis of ESCC include carcinoembryonic antigen (CEA), carbohydrate antigen 19-9 (CA19-9), cytokeratin 19 fragment (CYFRA21-1), neuron specific enolase (NSE), squamous cell carcinoma antigen (SCC), Dickkopf-related protein 1 (DKK1) and so on. CEA is mainly found in digestive system tissues. When the cells of the digestive system undergo malignant transformation, the serum CEA level generally increases significantly [55]. One study found that [56] in patients with ESCC who had undergone surgical treatment, the elevation of serum CEA during the postoperative reexamination had some value in predicting the increased risk of postoperative recurrence and distant metastasis, which could assist clinical diagnosis. The study of Qiao et al. [57] demonstrated that the upregulated SCC and CYFRA21-1 levels in ESCC patients before surgical treatment were associated with the invasive behavior of tumors. These two markers can be used to predict the prognosis of patients with ESCC. One study [58] tested multiple tumor markers in patients with esophageal cancer to evaluate the efficacy and prognosis of the patients. The authors found that the detection of the four markers combined had the highest diagnostic value and could effectively detect early tumor recurrence and metastasis. As reported in a retrospective study [59], the abnormal serum levels of CA19-9 and CYFRA21-1 in patients with esophageal cancer were associated with the occurrence and progression of esophageal cancer. Ju et al. showed that CYFRA21-1 and NSE played critical roles in the diagnosis and recurrence monitoring of esophageal cancer and had a diagnostic value when combined with CEA [60]. Kanda found that the optimal cutoff value of preoperative SCC-Ag concentrations for predicting ESCC recurrence was 1.1 ng/mL, which indicated that the serum SCC-Ag concentrations could be used as an easy monitoring tool for selecting a perioperative management method [61]. Other biomarkers such as Dickkopf-related protein 1 (DKK1) could serve as a novel biomarker for improving risk stratification and treatment monitoring of patients with esophageal cancer [62].

Based on the above findings, the diagnostic value of one serum marker may be low, but multiple biomarkers combined can significantly improve the sensitivity and specificity of diagnosis, and detect the recurrence and metastasis early of esophageal cancer after treatment.

#### 4.1.2. Tumor-Associated Antibodies

Tumor antigens can stimulate the immune system of the body to produce immune responses and generate antibodies. Tumor-associated antibodies (TAA) can mediate the dissolution and death of tumor cells. TAAs are produced in the stage of tumorigenesis and development. They are highly stable and easy to detect experimentally, and can be used as a potential tumor biomarker [63,64]. Therefore, screening for and detecting TAAs may help provide a strong basis for early diagnosis and detection of progression.

One study [65] evaluated the application value of 13 TAAs in detecting early-stage patients with ESCC and predicting cancer risk. The study found that serum p53 antibody was helpful to predict the prognosis of patients with esophageal cancer. Shimada et al. [66] reached a similar conclusion that serum p53 antibody in ESCC was superior to serological CEA, SCC and CYFRA21-1 biomarkers, and helpful to predict tumor recurrence and metastasis and monitor residual tumor cells. The researchers suggested that the lower the expression level of p53 in patients with esophageal cancer after surgery, the worse the prognosis. In a study by Heestand et al. [67], it was suggested that the level of anti-TOP48 antibody in the serum of patients with ESCC was significantly higher than those of patients with benign tumors and healthy controls. The ESCC patients with positive anti-TOP48 antibody had higher survival rate and longer survival time than those with negative antibody. The investigators concluded that anti-TOP48 antibody might be a serum biomarker for early diagnosis and prognosis of ESCC. One study [68] has shown that cytoskeleton-associated protein-4 (CKAP4) acts as a Dickkopf-1 (DKK-1) receptor in ESCC cells. The DKK1-CKAP4 pathway promotes cell proliferation and is therefore associated with poor prognosis and relapse-free survival. Anti-CKAP4 antibody can inhibit tumor formation by inhibiting the AKT activity, and can be used as a therapeutic target for ESCC. Heat shock protein-70 and heat shock protein-27 are associated with poor prognosis of ESCC and can be used as diagnostic and prognostic factors for patients [69]. One study found that 90% of 114 ESCC patients expressed peroxiredoxin 1 (Prdx1), and low expression of Prdx1 was associated with poor prognosis [70]. However, in a study by Zhang et al. [71], it was demonstrated that despite the abnormal expression of Prdx1, Prdx2 and Prdx6 in ESCC tissue samples, the peroxiredoxin subtype diversity was not associated with ESCC. Some researchers [72] observed that LY6K expression was found in 265 ESCC specimens, which confirmed that LY6K overexpression was significantly associated with poor prognosis of ESCC patients, and LY6K protein was found in the serum of ESCC patients with LY6K overexpression, which was related to the presence of tumors. In the study of Xu et al. [73], 513 participants were divided into two independent cohorts of 388 ESCC patients and 125 controls. The efficacy of p53, NY-ESO-1, MMP-7, heat shock protein-70, Prx VI and Bmi-1 antibodies in the diagnosis of ESCC was studied. The results showed that simultaneous detection of the six antibodies could differentiate early ESCC from normal controls. Other researchers proposed that simultaneous detection of antibodies against NY-ESO-1, STIP1 and MMP-7 had clinical value in the detection of early ESCC, but it was less effective in predicting the risk of ESCC [74]. According to the above research findings, simultaneous detection of multiple antibodies can obtain high diagnostic specificity and sensitivity, can detect early recurrence and metastasis, and has potential clinical application value.

#### 4.1.3. Other Blood Molecular Markers

With rapid research progress, it is found that blood circulating biomarkers play an important role in the diagnosis, treatment and evaluation of tumors. Among them, the appearance of circulating tumor cells has been regarded as one of the reasons for recurrence and metastasis in ESCC and is significantly associated with poor patient prognosis [75,76,77]. Circulating tumor cells, originating from cells in the primary tumor or metastasizing tumor, are released into the circulatory system after they are detached from the basement membrane and traverse the tissue, facilitating the formation of metastatic lesions in distant organs. Circulating tumor cells often appear in the peripheral circulation in different morphologies, such as circulating tumor DNA, exosomes as well as circulating cells. Circulating tumor DNA and cells carry all or part of the genetic information of the tumor, which can be used for gene and mutation detection to provide information for individualized treatment to improve the survival of the EC patients [78,79,80]. Exosomes are small vesicles released by cells under various physiological or pathological conditions and play a crucial role in intercellular communication by transferring various substances, such as proteins, lipids, nucleic acids and metabolites, to the receptor cells [81,82]. The formation of exosomes has an essential effect not only on the process of tumor development, but also on tumor diagnosis, treatment evaluation, dynamic monitoring and prognostic analysis [83]. A study has found that [84] exosome quantification can be used as an independent prognostic marker and can predict poor prognosis of patients with ESCC.

In addition, tumor cells secrete some other substances, such as miRNAs [85,86], lncRNAs [87] and circRNAs [88], which can be detected in blood and can be used for the diagnosis of ESCC by non-invasive methods. miRNAs are stable, abundant in expression and continuously detectable in blood, which has attracted the attention of researchers. A meta-study showed that the total sensitivity and specificity of miRNAs to detect ESCC were 79.9% and 81.3%, respectively [85]. Circulating miRNAs are clinically valuable biomarkers for ESCC; however, current studies are mostly limited to small clinical cohorts, and many studies focus on single miRNAs rather than combined applications. The development of bioinformatics has promoted the combined analysis of large and complex miRNA microarray data, which will provide sensitive and accurate biomarkers for early screening, diagnosis and treatment of esophageal cancer.

### 4.2. Molecular Cell Markers

Tumor cells release locally or into the circulation molecules that are unique and/or abnormal in concentration and could be used as tumor cell markers. Molecular cell markers play an essential role in the diagnosis, staging, response monitoring and recurrence detection of many cancers. Domestic and foreign scholars in basic research constantly find many cell markers with great clinical diagnostic and prognostic value in esophageal cancer. Numerous studies have found that cellular molecular markers are mainly oncogenes or tumor suppressor genes that are related to proliferation and metastasis, and mainly include DNA abnormal methylation, noncoding RNAs, transcription factors and so on.

Aberrant DNA methylation plays a crucial role in the carcinogenesis and development of tumors. The expression of abnormally methylated genes is often related to the degree of tumor differentiation, TNM stage and poor prognosis of ESCC patients. With the research progress, some abnormal methylation genes have begun to be used as markers for early diagnosis and prognosis prediction of tumors [89]. Pu et al. [90] analyzed and verified the role of methylation profiles of five genomic regions in the diagnosis and prognosis of ESCC by analyzing 100 samples from the high-throughput DNA methylation dataset of the Cancer Genome Atlas (TCGA) project and 12 samples from the comprehensive gene expression database (GEO). According to a study [91], methylation of HIN1, TFPI-2, DACH1 and SOX17 can be used as markers for early detection of esophageal cancer. Methylation of FHIT is related to poor prognosis of patients with ESCC, and methylation of CHFR is associated with chemotherapy sensitivity. This study proposed that aberrant DNA methylation can be used as a marker for the diagnosis, prediction of prognosis and chemosensitivity for esophageal cancer. In addition, researchers also found that the methylation levels of PAX1, SOX1 and ZNF582 in ESCC were higher than those in adjacent tissues, suggesting that the methylation status of PAX1, SOX1 and ZNF582 can be used as a potential biomarker for monitoring and diagnosis of ESCC [92,93]. Based on the above studies, exploring the characteristics of DNA methylation in ESCC is helpful to understand its mechanism and discover clinically valuable biomarkers.

Currently, with the development of high-throughput sequencing technology, differential gene expression profiling is a powerful technique to identify molecular markers of esophageal cancer phenotypes or predict prognosis [94]. Some studies have shown that noncoding RNAs [95], transcription factors [96] and other molecules play important roles in esophageal cancer, and they can be both tumor-suppressing and cancer-promoting molecules. Chen et al. [97] retrieved the miRNA expression profiles and clinical characteristics of patients with esophageal cancer from the TCGA database, and identified 18 miRNAs, six of which were associated with tumor recurrence and progression in patients with esophageal cancer. Mao et al. [98] identified seven lncRNAs that were associated with prognosis and constructed a prognostic prediction model by analyzing lncRNA expression microarray data of ESCC patients in the TCGA and GEO databases. The researchers found that these seven lncRNAs could be used as independent biomarkers for the prognosis prediction of patients with ESCC. The transcription factors LEF1, TEAD4, OCT4 and other molecules promote the biological behavior of esophageal cancer, and are associated with poor prognosis of esophageal cancer patients. They can be used as potential therapeutic targets and prognostic molecules for esophageal cancer [99,100,101]. According to the above analysis, molecular cell markers are of great clinical significance in the early diagnosis, treatment and prognosis evaluation of esophageal cancer, and can be used as potential novel tumor markers for esophageal cancer.

## 5. Postoperative Adjuvant Treatment Modalities

After surgery, the main purpose of adjuvant therapy for esophageal cancer patients is to kill the possible residual occult lesions and small metastases, so as to prevent recurrence and metastasis and improve the postoperative survival of patients. The main methods of postoperative adjuvant therapy are salvage surgery with minimally invasive procedure, radiotherapy, chemotherapy and immunotherapy.

### 5.1. Salvage Surgery

In a retrospective analysis by Itasu Ninomiya et al. [102], 128 patients who had undergone curative resection for esophageal cancer were included, of whom 37 patients developed recurrence after surgery. Of these patients who recurred, 29 had local therapy, including surgery in 10, surgery combined with postoperative radiotherapy in 2 and radiotherapy alone in 17 patients. The results of this study suggested that surgical treatment was associated with a better prognosis compared to other modalities in the initial treatment of recurrence, and the multivariate analysis revealed that treatment success at first recurrence was an independent variable in determining the prognosis after recurrence. In addition, Depypere et al. [103] reported similar results. Among patients with postoperative recurrence, the 5-year survival rate was 49.9% for those who underwent salvage surgery with or without adjuvant therapy, and 27% and 4.6% for chemoradiotherapy and chemotherapy alone, respectively. For patients with local recurrence, those whose lesions could be surgically removed based on multi-disciplinary consultation had prolonged survival time. If they could not be operated on, the combination of radiotherapy and chemotherapy seemed to provide a suboptimal choice. Nakamura et al. [104] retrospectively reviewed 68 patients with lymph node recurrence after radical resection of esophageal cancer, including 19 patients who received lymphadenectomy combined with adjuvant therapy, 22 who received radical radiotherapy and chemotherapy and 27 patients who received chemotherapy or the best supportive treatment. The 3-year survival rates of the lymphadenectomy plus adjuvant therapy group and the radical chemoradiotherapy group were 50.7% and 26.6%, respectively. There was no significant difference between the two groups, but the survival rates of the two groups were significantly higher than those of patients receiving chemotherapy alone or supportive care. 

In summary, salvage surgery is an essential treatment for local recurrence and metastasis of esophageal cancer after surgery at present, and timely intervention can benefit the survival of patients with recurrence. With the continuous development of minimally invasive techniques, surgery can be considered for most patients with esophageal cancer, owing to its advantages of minimal invasiveness and significant local control, which make it possible to perform salvage surgery for patients with postoperative recurrence and metastasis.

### 5.2. Chemoradiotherapy

At present, chemoradiotherapy is an effective and safe treatment for local recurrence of esophageal cancer patients, which can effectively control local tumor and improve the survival of patients. It is the main treatment for local recurrence after surgery. Radiotherapy has seen development from two-dimensional radiotherapy to three-dimensional conformal radiotherapy, intensity-modulated radiotherapy, image-guided radiotherapy, etc., with increased tumor targeting precision, which can improve the prognosis of patients with recurrence after esophageal cancer surgery and reduce the adverse effects of radiotherapy. Radiotherapy can control the local recurrence and metastasis of tumors, but its effect on tumor cells outside the target area is limited, and this is where chemotherapy can help. Therefore, the combination of radiotherapy and chemotherapy is a crucial treatment method for patients with recurrent esophageal cancer.

The retrospective analysis of Ma et al. [105] also found that for 98 patients with mediastinal recurrence after esophageal cancer surgery, the effective rates of synchronous radiotherapy and chemotherapy and radiotherapy alone were 91.8% and 73.5%, respectively, and the median survival times were 35 months and 19 months, respectively. The results demonstrated that synchronous radiotherapy and chemotherapy were superior to radiotherapy alone. Terufumi Kawamoto and other researchers [106] retrospectively analyzed the survival of 57 patients who had undergone radical resection of ESCC and received radiotherapy and chemotherapy after lymph node recurrence. The results suggested that about 28% of the patients had improved prognosis through radiotherapy and chemotherapy. In a retrospective study of Chen et al. [107], the data of 83 patients with local lymph node recurrence after radical resection of ESCC were analyzed. At the time of recurrence, 41 patients received radiotherapy alone and 42 patients received radiotherapy and chemotherapy. The results showed that the 3-year survival rate of the patients treated with radiotherapy alone was 47.5%, and the survival rate of the patients treated with radiotherapy combined with chemotherapy was 41.9%. The results of this retrospective study suggested that radiotherapy with or without chemotherapy was an effective treatment for local lymph node recurrence after radical resection of ESCC.

Based on the above findings, radiotherapy and chemotherapy are the main treatment measures for local recurrence of esophageal cancer patients. With continuous progress and development, precision radiotherapy can better protect adjacent normal organs and tissues. Chemotherapy alone is not the first choice for patients with local recurrence and metastasis after esophageal cancer surgery. It often needs to be combined with radiotherapy or surgery as part of comprehensive treatment to improve the survival of patients.

### 5.3. Immunotherapy

Immunotherapy is a hot topic of current clinical research, and with continuous development, it provides a new option for patients with postoperative recurrence of esophageal cancer. It has been documented that the high rate of positive PD-1/PD-L1 expression in ESCC has laid a solid foundation for immunotherapy in esophageal cancer [108,109].

In the ATTRACTION-3 study [108], which enrolled 419 patients with unresectable advanced or recurrent ESCC who were randomized at 1:1 to receive either nivolumab or chemotherapy, the overall survival was 10.5 months in the nivolumab group and 8 months in the chemotherapy group. Grade 3–4 treatment-related adverse events occurred in 18% of patients in the nivolumab group and 63% of patients in the chemotherapy group. The results of this study suggested that immunotherapy significantly improved overall survival and had a favorable safety profile compared with chemotherapy alone in patients with advanced ESCC. In a phase III study called Keynote 590 [110], the combination of pembrolizumab and chemotherapy was compared with chemotherapy alone in the treatment of patients with advanced esophageal cancer; the combined positive score (CPS) of PD-L1 was greater than or equal to 10, and a significant benefit in overall survival was observed in the pembrolizumab and chemotherapy group. An analysis from the Keynote 181 study [111] found that the addition of pabolizumab as a second-line treatment resulted in better prognosis than chemotherapy in patients with advanced esophageal cancer with a PD-L1 CPS score of 10 or more, and with fewer treatment-related adverse effects. The publication of the results of this study established pabolizumab as the second-line treatment for advanced ESCC. Some studies have demonstrated that chemoradiotherapy combined with immunotherapy can increase the expression of PD-L1 and enhance the ability of effector T cells to kill tumors and the abscopal effect of radiotherapy in patients with esophageal cancer [112]. Therefore, the combination of immunotherapy and chemoradiotherapy may further exert a synergistic effect to increase the disease control rate, thereby significantly improving survival rates. The CheckMate-577 phase III trial was to compare the effect of nivolumab monotherapy and placebo as a postoperative treatment for patients with preoperative chemoradiotherapy that have not achieved complete pathological response after complete resection of resectable esophageal cancer [113]. The use of nivolumab was to prevent recurrence and there is a therapy switch after recurrence in esophageal cancer patients. With continued research, immunotherapy is expected to change the traditional treatment pattern for patients with recurrent ESCC. 

## 6. Discussion

Esophageal cancer is one of the most common malignant tumors with high mortality in China. At present, the main treatment for early and middle clinical stage patients is comprehensive surgery-based treatment. However, the risk of recurrence and metastasis after surgery is high and the prognosis of patients is poor. For early detection of recurrence and metastasis after surgery, patients should be closely followed up with after treatment by means including serial clinical examinations, analysis of characteristic parameters of tumor and imaging and the use of biomarkers with high sensitivity and specificity. Monitoring the residual occult metastasis and tumor recurrence and timely treatment are expected to prolong the survival of patients. This article reviews the research progress of the factors influencing the recurrence of esophageal cancer from the clinical and basic research, hoping to provide new research directions and treatment strategies for clinical treatment and reexamination monitoring.

At present, many studies have been conducted to predict and prevent the recurrence and metastasis of esophageal cancer after surgery by exploring the related clinical factors, characteristic parameters predicted by imaging omics, and basic molecular markers, so as to achieve early detection and treatment and improve the prognosis of patients. However, most of the current studies are limited to small clinical cohorts, and many studies focus on single factors or molecules rather than combined application. There are certain limitations in using only one indicator to evaluate tumor occurrence. Therefore, to explore the influencing factors of postoperative recurrence of esophageal cancer, researchers should combine multiple indicators and expand the clinical cohort study to provide a sensitive and accurate prediction model for early detection, prognosis evaluation of recurrence and metastasis of esophageal cancer.

## Data Availability

Data sharing is not applicable to this article as no datasets were generated or analyzed during the current study.

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
