# Peer review of "Research Progress on the Predicting Factors and Coping Strategies for Postoperative Recurrence of Esophageal Cancer"

_cells, 2022, doi:10.3390/cells12010114_

Round 1
Reviewer 1 Report
1.Please supplement the latest literature contents and references of imaging omics and blood tumor markers
2、In the discussion section, please list the importance and timing of relevant factors for diagnosis, treatment and prognosis evaluation. That is, how to find and diagnose early and evaluate the treatment effect and prognosis early.
3.Language needs to be modified.
Reviewer 2 Report
This is an interesting narrative review on the topic.
However, i have some comments to make.
Lines 91-99: You refer to tumor length (instead of T stage) which may consist a prognostic factor for OS but does not belong to TNM, staging which is the heading of the paragraph. Please rephrase
Lines 99-103: You refer to vascular invasion which may consist a prognostic factor for OS but does not belong to TNM staging, which is the heading of the paragraph. Please rephrase.
Lines 151-154: Which are the sensitive detection methods for early diagnosis of recurrence you refer to ? Please define.
Lines 292-295: Are there clinical applications of multiple biomarkers combination for the early diagnosis of esophageal cancer recurrence ? Please add such evidence.
Lines 468-475: Unfortunately a study with only 7 patients cannot lead to any safe conclusions. Please ommit.
Finally, according to your aim " this article aims to review the research progress on the factors that infuence and predict postoperative recurrence of esophageal cancer". However, Section 4 refers to treatment modalities in cases of recurrence. Therefore, you should either remove it or expand the aim of the paper.
